# PermLLM: Learnable Channel Permutation for N:M Sparse Large Language Models

**Lancheng Zou**[1]**, Shuo Yin**[1]**, Zehua Pei**[1]**, Tsung-Yi Ho**[1]**, Farzan Farnia**[1]**, and Bei Yu**[1]

[1]The Chinese University of Hong Kong

## Abstract

Channel permutation is a powerful technique for enhancing the accuracy of N:M sparse models by reordering the channels of weight matrices to prioritize the retention of important weights. However, traditional channel permutation methods rely on handcrafted quality metrics, which often fail to accurately capture the true impact of pruning on model performance. To address this limitation, we propose PermLLM, a novel post-training pruning framework that introduces learnable channel permutation (LCP) for N:M sparsity. LCP leverages Sinkhorn normalization to transform discrete permutation matrices into differentiable soft permutation matrices, enabling end-to-end optimization. Additionally, PermLLM incorporates an efficient block-wise channel permutation strategy, which significantly reduces the number of learnable parameters and computational complexity. PermLLM seamlessly integrates with existing one-shot pruning methods to adaptively optimize channel permutations, effectively mitigating pruning-induced errors. Extensive experiments on the LLaMA series, Qwen, and OPT models demonstrate that PermLLM achieves superior performance in optimizing N:M sparse models. The code is available at `https://github.com/lanchengzou/PermLLM`.

## 1 Introduction

The rapid advancements in large language models (LLMs) [6, 61, 52, 1] have led to a notable enhancement in their capabilities across a broad range of domains. However, the growing scale of LLMs presents substantial challenges for efficient deployment. To address these challenges, model compression techniques, such as quantization [57, 16, 10, 31, 65] and pruning [15, 50, 62], offer promising solutions to reduce memory usage and computational overhead.

In this paper, we focus on network pruning [28, 22, 21], particularly semi-structured pruning [46, 42]. The core idea of network pruning is to eliminate redundancies within the model by preserving only the essential weights while setting the less important ones to zero. Semi-structured pruning takes this a step further by enforcing N:M sparsity, where N out of every M consecutive elements are set to zero. The N:M sparsity pattern is natively supported by Sparse Tensor Core in NVIDIA GPUs [45] to achieve speed-up, which makes semi-structured pruning a practical approach for efficient model inference.

Recent studies on LLM pruning primarily focus on designing a better pruning metric to obtain higher-quality masks to improve the accuracy of the sparse models [15, 50, 62]. RIA [62] introduces a novel pruning metric that avoids channel corruption while accounting for the effect of activations. Additionally, it proposes a two-stage channel permutation strategy to maximize the sum of retained weight importance, which serves as the quality metric to evaluate channel permutation solution. However, it is important to note that a discrepancy may exist between the handcrafted quality metric and the actual impact on output loss, as illustrated in Figure 1. Moreover, it fails to fully capture the

39th Conference on Neural Information Processing Systems (NeurIPS 2025).

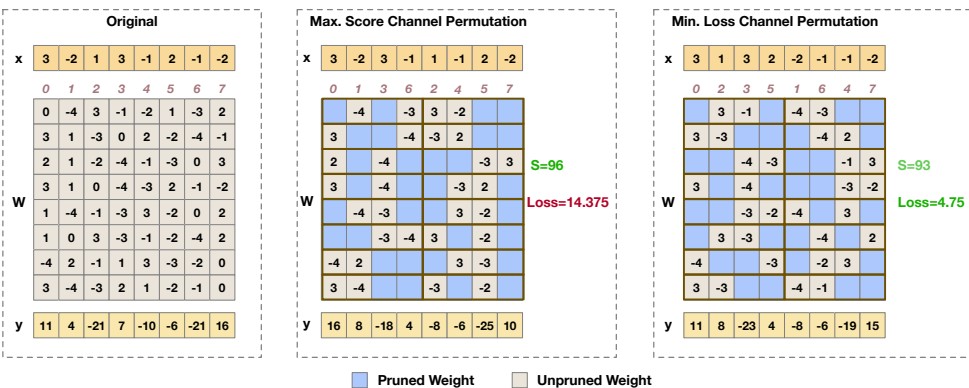

Figure 1: Effects of different channel permutation strategies on the outputs. Channel order is in **purple**. We use magnitude pruning [21] for 2:4 sparsity in this example. Score $S$ denotes the sum of retained weight importance, which is used as the quality metric for channel permutation [46, 62]. Loss is the mean square error between the original output **y** and the output of the pruned one. The output loss of direct 2:4 sparsity (i.e., without channel permutation) is 12.375. The results demonstrate that channel permutation which maximizes the score may lead to performance degradation.

complex inter-layer interactions, thereby missing opportunities to compensate for pruning errors and improve the overall performance of the sparse model.

To overcome the limitations of prior channel permutation methods, we are the first to present learnable channel permutation (LCP) for N:M sparsity. Unlike previous approaches that rely on handcrafted quality metrics as optimization proxies, the proposed post-training framework, PermLLM, directly minimizes the output errors between the dense model and the sparse model.

However, achieving feasible and practical permutation learning for pruning presents two major challenges: (1) the discrete nature and strict combinatorial constraints of permutation matrices render them non-differentiable, hindering effective optimization; (2) the vast solution space of permutations, particularly in LLMs with high-dimensional weight matrices, results in prohibitively high computational complexity.

To address these challenges, we first relax hard permutation matrices into soft permutation matrices using Sinkhorn normalization [48], enabling gradient-based optimization. Then, we introduce an efficient block-wise channel permutation strategy, which significantly reduces the number of learnable parameters and computational overhead. PermLLM is fully compatible with existing efficient one-shot pruning methods, such as Wanda [50] and RIA [62], enabling pruning-aware permutation learning that adaptively minimizes pruning-induced errors. Moreover, a customized CUDA kernel is developed to accelerate the channel permutation operation, achieving a significant speedup compared to the Pytorch implementation. Extensive experiments underscore the effectiveness of PermLLM, demonstrating its ability to enhance the performance of existing one-shot pruning methods across various LLMs, particularly for updated models such as LLaMA-3.1 and Qwen-2.5.

## 2 Preliminaries

### 2.1 Large Language Models Pruning

The effectiveness of network pruning [28, 22, 20, 21, 23] has garnered significant attention from researchers, prompting extensive exploration for LLM pruning.

Based on the granularity of pruning, prior works can be categorized into three types: structured pruning [36, 3, 49, 55, 38, 44], semi-structured pruning [62, 14] and unstructured pruning [15, 50, 4, 11]. Unstructured pruning is the most flexible approach, as it is not constrained by specific patterns. This flexibility often leads to improved accuracy; however, it comes at the expense of limited efficiency gains. In contrast, structured pruning removes weights at a coarse-grained level, such as channels [36], layers [38, 7], or blocks [49, 44], thereby enabling more substantial

improvements in computational efficiency. However, the structural removal often leads to significant accuracy degradation, necessitating retraining or fine-tuning to mitigate pruning-induced errors. Semi-structured pruning serves as an intermediate approach, introducing hardware-friendly patterns, such as N:M sparsity [42] which retains only N zero values within each group of M values. This method achieves a compromise between the acceleration benefits of structured pruning and the flexibility of fine-grained sparsity.

In general, there are three pipelines to obtain a sparse model [8]: pruning before training (PBT) [29, 54], pruning during training (PDP) [13, 32] and post-training pruning (PTP) [28, 22]. PBT and PDP typically demand substantial training efforts, which makes PTP the widely adopted pipeline for LLM pruning due to its lower computational cost. The objective of PTP can be formulated as follows:

$$\arg \min_{\mathbf{M}} \|\mathbf{W}\mathbf{X} - (\mathbf{M} \odot \mathbf{W}) \cdot \mathbf{X}\|_2^2, \quad \text{s.t. } \|\mathbf{M}\|_0 \leq k, \tag{1}$$

where $\mathbf{W} \in \mathbb{R}^{C_{out} \times C_{in}}$ represents the pre-trained weight with $C_{out}$ output channels and $C_{in}$ input channels. The goal of PTP is to determine a mask $\mathbf{M}$ that minimizes the reconstruction error under the given input $\mathbf{X}$ from calibration dataset and specific sparsity constraints (e.g., sparsity ratio and pruning granularity).

## 2.2 N:M Sparsity

NVIDIA Ampere architecture [45] leverages Sparse Tensor Core to accelerate model inference with N:M sparsity [42]. For instance, compressing the model with 2:4 sparsity can theoretically achieve a $2\times$ increase in compute throughput for sparse matrix multiplication compared to its dense counterpart. Thus, this approach has garnered significant attention for its ability to improve computational efficiency while maintaining model accuracy.

RIA [62] introduces a one-shot pruning method based on a handcrafted importance metric for semi-structured pruning. While one-shot pruning is highly efficient, it relies on handcrafted importance metrics as proxies for true discrepancy, resulting in a significant gap with the actual pruning-induced discrepancy. To address this issue, researchers have introduced various methodologies for learnable N:M masks [64, 34, 24, 26, 14]. Sparse-Refined Straight-Through Estimator (SR-STE) [64] is proposed by extending original Straight-Through Estimator (STE) [5] to train N:M sparse models from scratch.

## 2.3 Channel Permutation

Channel permutation [25, 46, 37] has proven to be an effective technique for improving the accuracy of pruning with specific sparsity patterns (e.g., N:M sparsity) by reordering the input channels of the weight matrix. More recently, researchers have explored reordering to enhance quantization performance [16, 59, 30], highlighting channel permutation as a promising approach that merits further investigation.

For a linear layer with $C_{in}$ input channels, there are $C_{in}!$ possible permutation candidates. Due to the nature of N:M sparsity, channel permutation can be formulated as the following problem: distributing $C_{in}$ distinguishable balls into $C_{in}/M$ indistinguishable boxes, where each box contains exactly $M$ balls. In this case, the solution space is reduced to $\frac{C_{in}!}{(M!)^G \cdot G!}$, where $G = C_{in}/M$ denotes the number of pruning groups. When $C_{in} = 16$ and $M = 4$, the reduced solution space still contains approximately 2.6 million candidates. The solution space grows rapidly with increasing $C_{in}$, leading to significant computational challenges for large values of $C_{in}$. Exhaustive search algorithm combined with a greedy incremental refinement strategy is applied for channel permutation [46]. However, this approach is primarily suitable for models with a small number of channels and becomes computationally expensive when applied to LLMs with large hidden dimensions. To address the computational overhead, RIA [62] adopts a heuristic channel allocation method that iteratively assign important channels to different blocks efficiently. Subsequently, a refinement process is applied, formulated as a linear sum assignment problem, to maximize the sum of retained weight importance scores.

Nevertheless, the handcrafted weight importance metric, used as a quality proxy in previous channel permutation methods [46, 62], fails to accurately capture the relationship between pruning error and channel permutation, resulting in suboptimal solutions. As illustrated in Figure 1, channel

permutation based on maximum importance score does not necessarily reduce pruning error and may even lead to an increase in error.

To address the aforementioned challenges and limitations, this study pushes the boundaries of post-training semi-structured pruning for LLMs by learnable channel permutation (LCP). This approach enables an end-to-end learning of channel reordering, eliminating the need for the handcrafted quality metrics. The proposed LCP serves as an effective plugin for existing one-shot pruning methods [50, 62] by identifying appropriate channel reordering to mitigate mask quality limitations and reduce pruning errors.

## 3 Learnable Channel Permutation

The objective of channel permutation is to determine a permutation matrix $\mathbf{P} \in \mathbb{R}^{C_{in} \times C_{in}}$ for the weight matrix $\mathbf{W} \in \mathbb{R}^{C_{out} \times C_{in}}$, such that the reordered weight matrix, $\widehat{\mathbf{W}} = \mathbf{WP}$, can achieve improved accuracy after applying N:M sparsity.

However, there are two major challenges to learn the permutation matrix $\mathbf{P}$: (1) $\mathbf{P}$ is a binary matrix containing only 0s and 1s, which makes it inherently discrete and thus non-differentiable. The discrete nature of $\mathbf{P}$ poses a significant challenge for gradient-based learning methods. Moreover, $\mathbf{P}$ must satisfy the properties of a permutation matrix—each row and column must contain exactly one "1" (with all other entries being "0"). This introduces strict combinatorial constraints that significantly increase the complexity of the learning process. (2) The number of possible permutation candidates increases factorially with $C_{in}$. In LLMs, $C_{in}$ typically exceeds one thousand, leading to an extremely vast solution space and posing a significant challenge for the design of efficient algorithms.

### 3.1 Relaxation to Soft Permutation Matrix

Some existing mask learning methods assign a learnable score [63] or probability [14] to each mask candidate to identify the best option. Although permutation learning can also be formulated as a combinatorial problem, the vast solution space of permutations renders these previously proposed methods impractical. Consequently, directly learning the permutation matrix tends to be more feasible.

To address the challenges associated with the discrete nature and properties of permutation matrix, a common approach is to relax the hard constraints and represent the permutation using a soft permutation matrix. The soft permutation matrix, denoted as $\widehat{\mathbf{P}}$, serves as a continuous and differentiable approximation of the discrete permutation matrix $\mathbf{P}$, thereby enabling gradient-based learning method. A **doubly stochastic matrix** can be used as $\widehat{\mathbf{P}}$ [2], where all entries are non-negative and each row and column sums to 1. This contrasts with $\mathbf{P}$, in which each row and column contains exactly one "1". By leveraging **Sinkhorn normalization** [48, 2, 39, 12, 35], a nonnegative square matrix can be converted into a doubly stochastic matrix through an iterative process of row and column normalization.

Thus, any square matrix $\mathbf{X}$ can be transformed into a doubly stochastic matrix as follows:

$$S^0(\mathbf{X}) = \exp(\mathbf{X}), \tag{2}$$

$$S^i(\mathbf{X}) = \mathcal{T}_c\big(\mathcal{T}_r(S^{i-1}(\mathbf{X}))\big), \tag{3}$$

$$S(\mathbf{X}) = \lim_{l \to \infty} S^l(\mathbf{X}), \tag{4}$$

where a non-negative square matrix is first obtained by Equation (2). Then iterative row and column normalization is performed by Equations (3) and (4). $\mathcal{T}_r(\mathbf{X}) = \mathbf{X} \oslash (\mathbf{X}\mathbf{1}_N\mathbf{1}_N^\top)$ is the row-wise normalization operation and $\mathcal{T}_c(\mathbf{X}) = \mathbf{X}\oslash(\mathbf{1}_N\mathbf{1}_N^\top\mathbf{X})$ is used for column normalization. $\oslash$ represents element-wise division and $\mathbf{1}_N$ denotes a column vector of one. Thus, the soft permutation matrix $\widehat{\mathbf{P}}$ can be obtained by

$$\widehat{\mathbf{P}} = S^L(\mathbf{W}_P/\tau), \tag{5}$$

where $\mathbf{W}_P$ is a learnable matrix with the same shape as $\widehat{\mathbf{P}}$. Since the limit in Equation (4) cannot be computed exactly in practice, a truncated version with $l \to L$ is typically used for implementation [39, 12]. The temperature coefficient $\tau$ controls the hardness of the soft permutation matrix: as $\tau$ approaches zero, the entries of $\widehat{\mathbf{P}}$ converge to either 0 or 1.

As $\widehat{\mathbf{P}}$ is not a strict permutation matrix, directly using it for channel permutation modifies both the channel order and the weight values. To avoid its impact on mask selection, $\widehat{\mathbf{P}}$ is hardened into a strict permutation matrix $\mathbf{P}$ during the forward pass. This hardening process can be formulated as a linear sum assignment problem and solved by using Hungarian algorithm [27]. Specifically, this process identifies the hard permutation matrix $\mathbf{P}$ that is closest to the soft permutation matrix $\widehat{\mathbf{P}}$. It achieves this by solving the following optimization problem:

$$\mathbf{P} = \arg\max_{\mathbf{P} \in \mathcal{P}} \mathrm{Tr}(\mathbf{P}^\top \widehat{\mathbf{P}}), \tag{6}$$

where $\mathcal{P}$ represents the set of all valid permutation matrices and $\mathrm{Tr}(\cdot)$ denotes the trace operator. The objective is to maximize the alignment between $\mathbf{P}$ and $\widehat{\mathbf{P}}$ by selecting the entries of $\widehat{\mathbf{P}}$ that yield the highest overall score. Unfortunately, the hardening process is not differentiable. To address this limitation, STE [5] is employed to approximate the gradient in the backward pass, i.e., $\partial\mathbf{P}/\partial\widehat{\mathbf{P}} = 1$. By propagating gradients through this approximation, the STE preserves gradient flow across the computational graph, thereby ensuring end-to-end trainability of the permutation learning framework.

## 3.2 Block-wise Learnable Channel Permutation

According to Equation (5), if channel is allowed to be permuted flexibly, the learnable parameter matrix will be $\mathbf{W}_P \in \mathbb{R}^{C_{in} \times C_{in}}$, which usually has the similar or same shape with the weight matrix $\mathbf{W}$. If each weight were to have its own learnable permutation, the learning burden would become prohibitively large.

To address this, we apply a block-wise learnable channel permutation that only allows channel permutation operates within the block to reduce the training cost. It is inspired by the widely adopted block-wise operations in model compression [16, 15, 66, 30].

Originally, the number of parameters in $\mathbf{W}_P$ is $C_{in}^2$ for full matrix learnable channel permutation. Let the block size be $B$. For permutation for a single block, the number of parameters in $\mathbf{W}_P^i$ is $B^2$. With $N_B$ representing the total number of blocks, the overall number of parameters is given by $N_B \times B^2 = \frac{C_{in}}{B} \times B^2 = C_{in} \times B$. By block-wise learnable channel permutation, we reduce the number of parameters to $\frac{B}{C_{in}}$ of the original, achieving significant parameter savings when $B \ll C_{in}$.

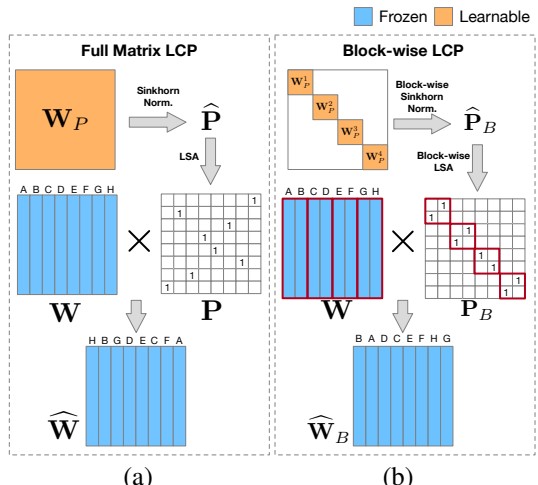

Figure 2: Illustration of learnable channel permutation with different granularity: (a) full matrix LCP; (b) block-wise LCP.

Another advantage of block-wise learnable channel permutation is its enhanced computational efficiency when hardening the soft permutation matrix. This process is solved using the Hungarian algorithm [27], which has a time complexity of $O(N^3)$. For a full matrix permutation, the time complexity becomes $O(C_{in}^3)$. In contrast, by adopting block-wise manner, the time complexity for a single block is $O(B^3)$. Given that there are $N_B$ blocks in total, the overall complexity is $O(N_B \cdot B^3) = O(C_{in} \cdot B^2)$. This demonstrates that it significantly reduces the computational cost of hardening process by utilizing block-wise learnable channel permutation, particularly when $B \ll C_{in}$.

To perform block-wise learnable channel permutation for $\mathbf{W}$, each learnable matrix $\mathbf{W}_P^i$ is transformed into a hard permutation matrix $\mathbf{P}_i$ for the $i$-th block. Unlike the reordered weight matrix $\widehat{\mathbf{W}} = \mathbf{W}\mathbf{P}$ obtained through full matrix permutation, the reordered weight matrix under block-wise permutation is given by $\widehat{\mathbf{W}}_B = \mathbf{W}\mathbf{P}_B$, where $\mathbf{P}_B = \mathrm{diag}(\mathbf{P}_1, \mathbf{P}_2, \ldots, \mathbf{P}_{N_B})$ represents a block diagonal matrix and $N_B$ is the number of blocks.

As illustrated in Figure 2, an example of block-wise learnable channel permutation with $N_B = 4$ is shown. In this case, the channels of $\mathbf{W}$ are partitioned into four blocks, with each block consisting of

consecutive $C_{in}/4$ channels. $\mathbf{P}_i$ only affects the channel permutation within the $i$-th block. Moreover, compared to full matrix permutation, only the diagonal blocks are learnable, while all other entries are fixed to zero, which significantly reduces the training overhead.

Given the advantages of the block-wise manner, it is adopted as the default setting for the proposed learnable channel permutation in the following sections. The full matrix approach can be considered a special case when the number of blocks is set to one.

## 4 PermLLM: Pruning with Learnable Channel Permutation

In this section, we will introduce the proposed novel N:M semi-structured pruning framework that combines the existing one-shot pruning methods [50, 62] with the proposed learnable channel permutation (LCP), which can further improve the performance of N:M sparse LLMs.

One-shot pruning eliminates weights by applying a predefined, handcrafted weight importance metric. For example, the weight importance metric proposed by Wanda [50] is defined as $\mathbf{S}_{ij} = |\mathbf{W}_{ij}| \cdot ||\mathbf{X}_j||_2$, where $\mathbf{W} \in \mathbb{R}^{C_{out} \times C_{in}}$ and $\mathbf{X}$ is the input from calibration. Subsequently, the pruning mask $\mathbf{M} \in \mathbb{R}^{C_{out} \times C_{in}}$ is determined to maximize the sum of the retained importance metrics, which can be formulated as

$$\arg\max_{\mathbf{M}} \sum_{i=0}^{C_{out}} \sum_{k=0}^{C_{in}/M} \sum (\mathbf{M} \odot \mathbf{S})_{i,kM:(k+1)M}, \quad \text{s.t.} \ ||\mathbf{M}_{i,kM:(k+1)M}||_0 = M - N, \quad (7)$$

where $\mathbf{M}_{i,kM:(k+1)M}$ is constructed by setting the entries corresponding to the largest $M - N$ values in $\mathbf{S}_{i,kM:(k+1)M}$ to 1, while all other entries are set to 0. This approach achieves N:M sparsity by ensuring that $N$ out of every $M$ consecutive elements are set to 0, while preserving the most important weights based on their importance metrics.

With channel permutation, the order of the channels is rearranged, and consequently, the channels in the importance matrix $\mathbf{S}$ are permuted accordingly. The permuted importance matrix is represented as $\widehat{\mathbf{S}} = \mathbf{S}\mathbf{P}_B$, where $\mathbf{P}_B$ is the permutation matrix. As a result, the mask $\mathbf{M}$ varies depending on the specific permutation solution for the permuted weight $\widehat{\mathbf{W}} = \mathbf{W}\mathbf{P}_B$:

$$\arg\max_{\mathbf{M}} \sum_{i=0}^{C_{out}} \sum_{k=0}^{C_{in}/M} \sum (\mathbf{M} \odot \widehat{\mathbf{S}})_{i,kM:(k+1)M}, \quad \text{s.t.} \ ||\mathbf{M}_{i,kM:(k+1)M}||_0 = M - N. \quad (8)$$

However, the non-differentiability of the `argmax` operation hinders gradient backpropagation, rendering it unsuitable for gradient-based learning frameworks. To address this, STE [5] is employed to approximate the gradients during the backward pass. Specifically, while the forward pass uses the non-differentiable `argmax` operation to obtain a discrete hard mask $\mathbf{M}$, the backward pass introduces a soft mask $\widehat{\mathbf{M}}$ to enable gradient computation. The soft mask is defined as:

$$\widehat{\mathbf{M}}_{i,kM:(k+1)M} = \text{Softmax}(\widehat{\mathbf{S}}_{i,kM:(k+1)M}), \quad (9)$$

where the `softmax` function provides a continuous and differentiable approximation. This approach allows the forward pass to retain the discrete selection behavior of `argmax`, while the backward pass leverages the smooth and differentiable properties of `softmax` to compute gradients effectively.

Existing channel permutation methods [46, 62] are primarily designed to find the optimal permutation matrix $\mathbf{P}^*$ and the corresponding mask $\mathbf{M}^*$ that maximize the sum of retained importance score, as defined in Equation (8). However, the handcrafted quality metric used to evaluate channel permutation solutions often fails to accurately reflect the true effectiveness of the permutation, potentially leading to suboptimal outcomes or even worse performance as illustrated in Figure 1.

To address the aforementioned issue, PermLLM aims to directly minimize the output discrepancy between the dense model and the sparse N:M model by incorporating learnable channel permutations. Specifically, we utilize a cosine similarity loss to encourage alignment between the outputs of the two models, which is defined as:

$$\mathcal{L}_{cosine}(\mathbf{y}, \widetilde{\mathbf{y}}) = 1 - \frac{\mathbf{y} \cdot \widetilde{\mathbf{y}}}{||\mathbf{y}|| \cdot ||\widetilde{\mathbf{y}}||}, \quad (10)$$

where $\mathbf{y}$ and $\widetilde{\mathbf{y}}$ represent the outputs of the original dense model and the sparse N:M model.

During the proposed post-training pruning process, only $\mathbf{W}_P^i$ for each permutation matrix $\mathbf{P}_B$ is learnable, while all weight matrices remain fixed, as illustrated in Figure 2. Additionally, each mask $\mathbf{M}$ is directly obtained from Equation (8), with its values dynamically updated based on changes in $\mathbf{P}_B$. By leveraging the proposed relaxation and gradient approximation techniques, the optimization of each $\mathbf{P}_B$ is effectively guided toward solutions that maximize the preservation of the dense model's performance while adhering to the N:M sparsity constraint.

After training, weight $\mathbf{W}$ will be permuted and pruned by

$$\widehat{\mathbf{W}}' = \mathbf{M}^* \odot (\mathbf{W}\mathbf{P}_B^*), \tag{11}$$

where $\mathbf{P}_B^*$ denotes the learned channel permutation matrix and $\mathbf{M}^*$ is the corresponding pruning mask.

Notably, the channels of the input activations must also be permuted to align with the channel order of the weight matrix. It can be accomplished by permuting the output channels of the preceding layer. Let $\mathbf{P}_{l,B}^*$ denote the permutation matrix for the current layer, and let $\widehat{\mathbf{W}}_{l-1}'$ represent the permuted and pruned weight matrix of the preceding layer. The row of $\widehat{\mathbf{W}}_{l-1}'$ should be reordered for input activation permutation of its succeeding layer, which can be expressed as:

$$\widehat{\mathbf{W}}_{l-1}'' = \mathbf{P}_{l,B}^* \widehat{\mathbf{W}}_{l-1}'. \tag{12}$$

Since it is a row-wise operation, it preserves the N:M sparsity of $\widehat{\mathbf{W}}_{l-1}'$. To further reduce the runtime overhead introduced by channel permutations, we developed a customized CUDA kernel specifically for the channel permutation operation. Experimental results evaluated on LLaMA-2 7B demonstrate that this kernel achieves an average speedup of $84\times$ compared to the Pytorch implementation, thereby making pruning with channel permutations significantly more practical.

## 5  Experiments

### 5.1  Setups

We compare with three baselines in N:M sparsity, especially 2:4 sparsity: SparseGPT [15], Wanda [50] and RIA [62]. Wanda/RIA-CP enables channel permutation for N:M sparsity introduced in RIA. PermLLM$_{Wanda/RIA}$ indicates that Wanda or RIA is employed as the pruning metric in our PermLLM framework.

The proposed method is evaluated on various open source representative models: LLaMA 7B-13B [51], LLaMA-2 7B-13B [52], LLaMA-3.1 8B [19], Qwen-2.5 7B [58], and OPT 6.7B [61]. We randomly select 128 samples from the C4 dataset [47], each comprising 1024 tokens, to serve as the calibration data for all evaluated models. We utilize five zero-shot evaluation tasks: HellaSwag [60], ARC-(Easy and Challenge) [9], OpenBookQA [41] and RTE [53] from lm-evaluation-harness [18] and one language modeling dataset: Wikitext2 [40] to evaluate the performance of the sparse models.

We implement PermLLM with Pytorch [43] and HuggingFace Transformers library [56]. The experiments of PermLLM are conducted on A100 GPUs. We employ N:M semi-structured pruning for linear layers, skipping the initial embedding layer and the final classification head. These linear layers constitute approximately 99% of the total parameters in LLMs.

For the proposed PermLLM framework, we utilize AdamW [33] as the optimizer, with the learning rate set from {1e-3, 5e-3} for all models. The iteration of Sinkhorn normalization is 5. The temperature $\tau$ is linearly decayed from 1 to 0.1 to control the hardness of the soft permutation matrix in Equation (5). The block size for block-wise learnable channel permutation is set to 64, as it offers a balanced trade-off between performance and efficiency. Specifically, a block size of 64 is considered a more practical choice, as increasing the block size to 128 results in a twofold increase in runtime. This is because a larger block size not only raises computational complexity but also requires more iterations to achieve convergence due to the significantly expanded solution space. The pruning duration is about 2.5 hours for the 7B model with 4 GPUs and 5.5 hours for the 13B model with 8 GPUs, which is considered acceptable given the extremely large-scale nature of pruning-aware permutation problem. More efficient implementation scheme of PermLLM is discussed in Appendix A.

Table 1: 2:4 semi-structured pruning results on Wikitext2 with perplexity as the evaluation metric.

| Method | OPT 6.7B | LLaMA 7B | LLaMA 13B | LLaMA-2 7B | LLaMA-2 13B | LLaMA-3.1 8B | Qwen-2.5 7B |
|---|---|---|---|---|---|---|---|
| Dense | 10.86 | 5.68 | 5.09 | 5.47 | 4.89 | 6.24 | 7.74 |
| SparseGPT | 14.33 | 11.19 | 9.17 | 11.12 | 9.03 | 16.62 | 14.34 |
| Wanda | 16.29 | 11.59 | 9.60 | 12.16 | 9.05 | 23.42 | 24.44 |
| Wanda+CP | 15.28 | 11.07 | 8.69 | 11.00 | 8.51 | 21.09 | 18.76 |
| PermLLM$_{Wanda}$ | 14.27 | **9.41** | 8.06 | **9.39** | 8.20 | **14.03** | **13.58** |
| RIA | 15.93 | 11.14 | 8.96 | 11.30 | 8.51 | 22.62 | 22.67 |
| RIA+CP | 15.13 | 10.99 | 8.15 | 10.26 | 8.08 | 19.80 | 17.58 |
| PermLLM$_{RIA}$ | **14.23** | 9.95 | **7.81** | 9.60 | **7.97** | 15.79 | 15.93 |

Table 2: Zero-shot performance of 2:4 sparse models.

| Model | Method | Weight Update | HellaSwag | ARC_E | ARC_C | OBQA | RTE | Average |
|---|---|---|---|---|---|---|---|---|
| OPT 6.7B | Dense | - | 50.46 | 65.49 | 30.12 | 26.80 | 55.23 | 45.62 |
| | SparseGPT | ✓ | 43.40 | **60.82** | 26.62 | **24.40** | 52.71 | 41.59 |
| | Wanda | ✗ | 41.56 | 57.62 | 24.83 | 23.00 | 53.43 | 40.09 |
| | Wanda+CP | ✗ | 42.87 | 59.51 | 26.02 | 22.00 | 52.71 | 40.62 |
| | PermLLM$_{Wanda}$ | ✗ | **44.27** | 59.43 | **27.22** | 24.00 | **54.15** | **41.81** |
| LLaMA 7B | Dense | - | 56.95 | 75.38 | 41.89 | 34.80 | 65.34 | 54.87 |
| | SparseGPT | ✓ | 43.55 | 61.78 | 27.90 | 22.80 | 58.12 | 42.83 |
| | Wanda | ✗ | 42.33 | 61.57 | 28.07 | 23.60 | 51.26 | 41.37 |
| | Wanda+CP | ✗ | 44.21 | 63.51 | 29.86 | 24.00 | 58.12 | 43.94 |
| | PermLLM$_{Wanda}$ | ✗ | **47.03** | **63.30** | **30.55** | **25.00** | **62.45** | **45.67** |
| LLaMA-2 7B | Dense | - | 57.13 | 76.30 | 43.26 | 31.60 | 62.45 | 54.15 |
| | SparseGPT | ✓ | 44.11 | 64.14 | **31.31** | 24.20 | 58.84 | 44.52 |
| | Wanda | ✗ | 41.59 | 61.74 | 30.20 | 24.00 | 53.07 | 42.12 |
| | Wanda+CP | ✗ | 43.40 | 64.69 | 30.03 | 26.00 | 53.07 | 43.44 |
| | PermLLM$_{Wanda}$ | ✗ | **46.60** | **65.49** | 31.14 | **26.20** | **63.54** | **46.59** |
| LLaMA-3.1 8B | Dense | - | 60.06 | 81.48 | 51.28 | 33.40 | 70.04 | 59.25 |
| | SparseGPT | ✓ | 44.25 | **63.76** | 30.55 | **24.20** | 53.79 | 43.31 |
| | Wanda | ✗ | 38.45 | 58.00 | 26.37 | 19.40 | 52.35 | 38.91 |
| | Wanda+CP | ✗ | 39.32 | 62.25 | 28.92 | 20.40 | 52.71 | 40.72 |
| | PermLLM$_{Wanda}$ | ✗ | **45.33** | 62.58 | **30.97** | 24.00 | 53.79 | **43.33** |
| Qwen-2.5 7B | Dense | - | 58.79 | 79.56 | 46.08 | 33.00 | 76.90 | 58.87 |
| | SparseGPT | ✓ | 46.20 | **71.13** | 37.46 | 26.00 | 75.45 | 51.25 |
| | Wanda | ✗ | 40.60 | 67.17 | 33.45 | 25.40 | 72.92 | 47.91 |
| | Wanda+CP | ✗ | 42.92 | 70.50 | 36.09 | 25.20 | 72.20 | 49.38 |
| | PermLLM$_{Wanda}$ | ✗ | **47.30** | 70.58 | **38.13** | **27.60** | **77.26** | **52.17** |

## 5.2 N:M Semi-structured Pruning for LLMs

**Language Modeling.** In Table 1, we evaluate the language modeling performance of the 2:4 sparse models on Wikitext2. Perplexity is used as the evaluation metric, with lower values indicating better language modeling performance. SparseGPT updates the remaining unpruned weights during pruning to compensate the pruning error. Other pruning methods, including our proposed PermLLM, do not modify weight values.

Empirical results demonstrate that channel permutations effectively mitigate performance degradation in pruned models. However, existing channel permutation algorithms rely on handcrafted heuristic metrics to generate permutations, often yielding suboptimal solutions. In contrast, PermLLM employs end-to-end learnable optimization to derive superior permutations by directly minimizing the performance gap between the dense and pruned models. Compared to SparseGPT, both Wanda and RIA initially demonstrate superior performance on LLaMA and LLaMA-2. The proposed PermLLM framework further unlocks their potential. On the other hand, for other models, Wanda and RIA underperform relative to SparseGPT, even with channel permutations. Specifically, significant performance degradations are observed in LLaMA-3.1 and Qwen-2.5 even using Wanda+CP and RIA+CP. However, with the incorporation of learnable channel permutations, PermLLM surpasses

Table 3: Runtime for the different layers and channel permutations in LLaMA-2 7B using 2048 tokens.

| Method | Q/K/V/O_proj | Up/Gate_proj | Down_proj | CP |
|--------|--------------|--------------|-----------|-----|
| Dense | 1.513ms | 2.607ms | 2.614ms | - |
| 2:4 sparsity + CP | 0.927ms | 1.526ms | 1.535ms | 0.039ms |
| Speedup | 1.632× | 1.708× | 1.703× | - |

Table 4: Evaluation on PermLLM$_{wanda}$ for LLaMA-2 7B with different iteration number of Sinkhorn normalization.

| Model | # of Iter. | HellaSwag | ARC_E | ARC_C | OBQA | RTE | Average | Wikitext2 |
|-------|-----------|-----------|-------|-------|------|-----|---------|-----------|
| Qwen-2.5 7B | 0 | 45.28 | **64.65** | 29.86 | 21.20 | **53.79** | 42.96 | 14.12 |
|  | 5 | **45.33** | 62.58 | **30.97** | **24.00** | **53.79** | **43.33** | **14.03** |
| LLaMA-3.1 8B | 0 | 45.93 | **71.00** | 37.88 | 25.40 | 65.70 | 49.18 | 14.43 |
|  | 5 | **47.30** | 70.58 | **38.13** | **27.60** | **77.26** | **52.17** | **13.58** |

SparseGPT due to its accurate and model-wise optimization, demonstrating the effectiveness and superiority of the proposed framework.

**Zero-shot Performance.** In Table 2, we report the zero-shot performance of the 2:4 sparse models on five evaluation tasks. The average accuracy across all tasks is presented in the last column. PermLLM significantly enhances the effectiveness of channel permutations for pruning, outperforming existing methods on the majority of tasks and achieving the highest average accuracy. This highlights the significant potential of channel permutation as an effective tool for semi-structured pruning. We also evaluate PermLLM for 4:8 sparsity on LLaMA-2 7B in Table 8, which shows PermLLM is not limited to 2:4 sparsity.

**Inference Speedup**. Inference runtime evaluation is crucial to validate the practicability of the proposed framework. In Table 3, we report the runtime speedup of 2:4 sparse LLaMA-2 model using a batch of 2048 tokens following SparseGPT and RIA. The customized CUDA kernel of channel permutation reduces the total runtime from 3.288ms to 0.039ms, providing 84× speedup compared to Pytorch implementation. Thus, the overhead of channel permutations is minimal with the customized CUDA kernel. The overall acceleration across all linear layers, even with channel permutations, is approximately 1.67×.

### 5.3 Ablation Study

We conduct an ablation study on the relaxation of the soft permutation matrix to evaluate its impact on our framework. A larger iteration number in Sinkhorn normalization allows the soft permutation matrix to converge more closely to a doubly stochastic matrix (DSM). By default, we set the iteration number of Sinkhorn normalization to 5, which generally yields satisfactory performance. In Table 4, we evaluate the pruning performance of PermLLM$_{Wanda}$ under different Sinkhorn normalization iterations. When the iteration number is set to 0, the soft permutation matrix deviates the most from a DSM. The results demonstrate the benefits of using a DSM as the soft permutation matrix for permutation learning. It helps to enhance the learning process by providing a more structured and meaningful representation.

In Table 5, we evaluate PermLLM$_{wanda}$ on the LLaMA-2 7B model using different calibration datasets: Pile [17], Wikitext2 [40], and C4 [47]. Each dataset consists of 128 randomly selected samples. The results demonstrate that the learned permutation performs consistently well across different datasets, which indicates robustness of PermLLM.

Additionally, we conduct experiments to analyze the trade-off between performance and training cost under varying block sizes. As shown in Table 6, larger block sizes provide a greater optimization space. However, this increased space comes at the cost of longer exploration and convergence times. We select a block size of 64 as the default, as it strikes a good balance between pruning performance and training efficiency.

Table 5: Evaluation on PermLLM$_{wanda}$ for LlaMA-2 7B with different calibration dataset.

| Dataset | HellaSwag | ARC_E | ARC_C | OBQA | RTE | Average | Wikitext2 |
|---------|-----------|-------|-------|------|-----|---------|-----------|
| Pile | 45.83 | 64.31 | 32.08 | 26.60 | 54.87 | 44.74 | 8.96 |
| Wikitext2 | 45.42 | 66.41 | 32.34 | 25.80 | 53.07 | 44.61 | 8.31 |
| C4 | 46.60 | 65.49 | 31.14 | 26.20 | 63.54 | 46.59 | 9.39 |

Table 6: Evaluation on PermLLM$_{wanda}$ for LlaMA-2 7B with different block size.

| Block size | HellaSwag | ARC_E | ARC_C | OBQA | RTE | Average | Wikitext2 | Time |
|------------|-----------|-------|-------|------|-----|---------|-----------|------|
| 32 | 46.13 | 64.39 | 29.69 | 24.60 | 53.07 | 43.58 | 9.50 | 2h |
| 64 | 46.60 | 65.49 | 31.14 | 26.20 | 63.54 | 46.59 | 9.39 | 2.5h |
| 128 | 46.47 | 66.08 | 32.08 | 27.40 | 64.43 | 47.09 | 9.07 | 6h |

## 6 Conclusion

This paper introduces PermLLM, a novel pruning framework leveraging learnable channel permutations (LCP) to optimize N:M sparsity in large language models. By minimizing pruning errors through end-to-end optimization, PermLLM significantly enhances the performance of N:M semi-structured pruning. Experimental results validate its superiority over existing methods.

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

# A    Implementations

**Hyperparameters Configurations.** The learning rate is set from {1e-3, 5e-3}. Specifically, we use 1e-3 for PermLLM$_{Wanda}$ and 5e-3 for PermLLM$_{RIA}$. The iteration of Sinkhorn normalization is 5. The temperature $\tau$ is linearly decayed from 1 to 0.1 to control the hardness of the soft permutation matrix in Equation (5). The block size for block-wise learnable channel permutation is set to 64, as it offers a balanced trade-off between performance and efficiency. We use 50 iterations for permutation learning.

**More Efficient Implementation.** PermLLM serves as an effective plugin to improve performance of existing zero-shot pruning methods. As observed in other studies, different layers have varying impacts on the output. To further enhance the efficiency of PermLLM, learnable channel permutation modules can be inserted into only a subset of layers, while the traditional channel permutation method is applied to the remaining layers. For instance, we apply learnable channel permutations only to the last six decoder layers of the LLaMA-2-7B model. In this case, only a single GPU is required for permutation learning, reducing the runtime to 0.4 hours, which is similar to the runtime of traditional channel permutation method.

The experimental results are shown in Table 7. Although partial PermLLM does not match the performance of full PermLLM due to its limited optimization space, it still provides notable improvements over traditional channel permutation methods. This approach also represents a balanced trade-off between performance and efficiency, making it particularly suitable for scenarios with relatively limited computational resources.

Table 7: Experimental results on LLaMA-2-7B with partial PermLLM. We highlight the top-2 results.

| Method | HellaSwag | ARC_E | ARC_C | OBQA | RTE | Average | Wikitext2 |
|---|---|---|---|---|---|---|---|
| RIA+CP | 42.86 | **64.69** | 30.29 | 24.40 | **54.87** | 43.42 | 10.26 |
| PermLLM$_{RIA}$ (partial) | **44.46** | 64.10 | **31.74** | **24.80** | 53.79 | **43.78** | **10.10** |
| PermLLM$_{RIA}$ (full) | **45.15** | **64.77** | **32.25** | **24.80** | 54.51 | **44.30** | **9.60** |

# B    PermLLM for 4:8 Sparsity

In Table 8, we present the detailed results about the 4:8 sparsity on LLaMA-2 model with different pruning methods. The experimental results demonstrate that PermLLM is not limited to 2:4 sparsity and can still outperform traditional method for 4:8 sparsity.

Table 8: Evaluation on 4:8 sparse LLaMA-2-7B with different pruning methods.

| Method | Weight Update | HellaSwag | ARC_E | ARC_C | OBQA | RTE | Average | Wikitext2 |
|---|---|---|---|---|---|---|---|---|
| Dense | - | 57.13 | 76.30 | 43.26 | 31.60 | 62.45 | 54.15 | 5.47 |
| SparseGPT | ✓ | 48.77 | 67.68 | 34.81 | 26.20 | 53.79 | 46.25 | 8.56 |
| Wanda | ✗ | 46.87 | 66.92 | 34.04 | 26.40 | 54.87 | 45.82 | 8.63 |
| Wanda+CP | ✗ | 48.61 | **70.62** | 35.15 | 28.60 | **55.23** | 47.64 | 8.26 |
| PermLLM$_{Wanda}$ | ✗ | **49.02** | 70.20 | **36.35** | **29.40** | 54.87 | **47.97** | **7.96** |

# C    Visualization of Mask

Figure 3 illustrates the masks of layer.30.down_proj in the pruned LLaMA-2-7B by different methods. For methods involving channel permutations (e.g., RIA+CP and PermLLM$_{RIA}$), the channels are permuted back to their original order for better comparison. We extract a 128×128 portion of the mask (i.e., mask[:128, :128]) for better visualization. It's observed that the retained weights differ between the previous channel permutation method and our proposed learnable channel permutation. This is because we utilize different strategies: previous one aims at maximizing the sum of the retained importance metrics and PermLLM is to minimize the output discrepancy between dense model and the pruned model.

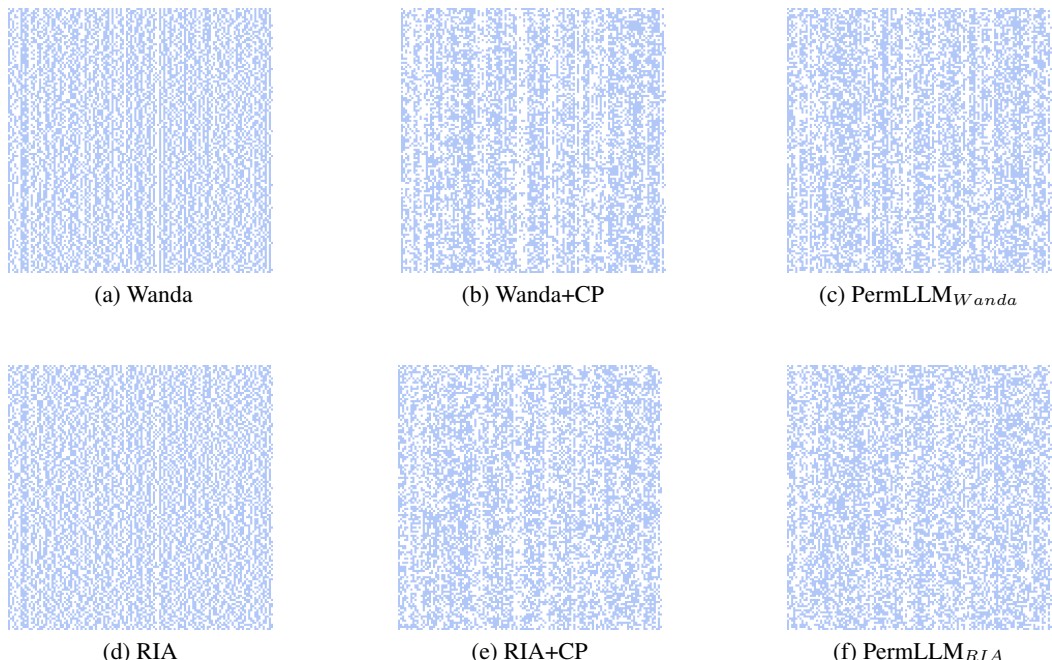

Figure 3: Visualization of mask obtained by different pruning methods. The blue part means the pruned weights and the white part is the retained weights

## D   Limitations

This paper introduces a innovative learnable channel permutation method to enhance semi-structured pruning for the first time. Although the method is tailored for semi-structured pruning, channel permutation or channel reordering has also been shown to be beneficial in other areas, such as quantization [30, 59]. This suggests that the broader applicability of the proposed approach to tasks beyond pruning, such as optimizing quantization performance, remains an open area for future exploration. Moreover, while the proposed block-wise channel permutation scheme significantly reduces training overhead compared to the full matrix scheme, the training of PermLLM still requires more computational resources compared to traditional channel permutation methods. Enhancing the training efficiency for pruning-aware permutation learning remains an important direction for future research.

## E   Broader Impacts

The proposed learnable channel permutation for N:M sparsity is not expected to have any negative societal impacts. Instead, it has the potential to advance the field of machine learning, particularly in the area of model compression.

