# OpenReview forum: "PermLLM: Learnable Channel Permutation for N:M Sparse Large Language Models"
_NeurIPS.cc/2025/Conference — NeurIPS 2025 poster_

### Official Review · Reviewer_fovA · 2025-06-18

**Clarity:** 3
**Significance:** 2
**Originality:** 3
**Rating:** 4
**Confidence:** 4

**Summary:**

The paper proposes PermLLM, a novel framework for N:M sparse pruning of large language models (LLMs), introducing learnable channel permutation (LCP) to optimize sparsity and improve model performance post-pruning.

**Questions:**

1.All evaluations are performed under 50% sparsity (2:4 or 4:8). Could the performance be further compared under different sparsity levels, such as 1:4 and 3:4?

2.The experimental results in Table 2 are incomplete. A more comprehensive comparison using different methods on the same model should be conducted.

3.It would be beneficial to include some of the latest unstructured pruning methods, such as Pruner-Zero [1] and MaskLLM [2].

[1] Dong P, Li L, Tang Z, et al. Pruner-zero: Evolving symbolic pruning metric from scratch for large language models. arXiv preprint arXiv:2406.02924, 2024.

[2] Fang G, Yin H, Muralidharan S, et al. MaskLLM: Learnable semi-structured sparsity for large language models. arXiv preprint arXiv:2409.17481, 2024.

**Ethical Concerns:**

["NO or VERY MINOR ethics concerns only"]

**Final Justification:**

The newly added experiments provide a more comprehensive demonstration of the method's effectiveness, so I have raised my score to 4.

**Limitations:**

Yes

**Quality:**

3

**Strengths And Weaknesses:**

**Strengths**:

1.The method can seamlessly integrate with existing one-shot pruning techniques like Wanda and RIA, extending their capabilities with adaptive channel optimization.

2.Extensive experiments across multiple models (LLaMA, Qwen, OPT) demonstrate the effectiveness of PermLLM in improving pruning results.

**Weaknesses**:

1.The experimental scope is limited, as only 2:4 semi-structured pruning experiments were conducted, which restricts its generalizability.

---

> ### Author Rebuttal · Authors · 2025-07-31
>
> Thanks for your valuable feedback.
>
> **Q1**: We conduct extra experiments on LLaMA-2 7B under 1:4 sparsity.
>
> |          Method          | HellaSwag | ARC_E     |   ARC_C   |   OBQA    |    RTE    |  Average  | Wikitext2 |
> | :----------------------: | :-------: | --------- | :-------: | :-------: | :-------: | :-------: | :-------: |
> |          Wanda           |   55.13   | 75.08     |   42.83   |   30.60   | **58.12** |   52.35   |   5.91    |
> |         Wanda+CP         |   55.56   | 75.38     |   42.06   |   32.00   |   55.70   |   52.14   |   5.87    |
> | $\text{PermLLM}_{Wanda}$ | **55.79** | **75.97** | **43.34** | **32.80** |   56.32   | **52.84** | **5.81**  |
>
> We also conduct extra experiments on LLaMA-2 7B under 3:4 sparsity.
>
> |          Method          | HellaSwag | ARC_E     |   ARC_C   |   OBQA    |    RTE    |  Average  | Wikitext2  |
> | :----------------------: | :-------: | --------- | :-------: | :-------: | :-------: | :-------: | :--------: |
> |          Wanda           |   26.25   | 26.39     |   19.54   |   11.80   |   52.71   |   27.61   |  2041.50   |
> |         Wanda+CP         |   26.05   | 27.53     | **20.48** | **14.60** |   52.71   |   28.27   |  2128.15   |
> | $\text{PermLLM}_{Wanda}$ | **26.42** | **29.08** |   20.31   | **14.60** | **53.07** | **28.70** | **744.38** |
>
> The results demonstrates that PermLLM can further improve the pruning performance under 1:4 and 3:4 sparsity.
>
> **Q2**: In Table 2, we list the better results between $\text{PermLLM}_{Wanda}$ and $\text{PermLLM}_{RIA}$ due to the page limitation.
>
> Here, we list more detailed results.  And we will list the complete results in the final version.
>
> For LLaMA-3.1 8B,
>
> |         Method         | HellaSwag | ARC_E     |   ARC_C   |   OBQA    |    RTE    |  Average  |
> | :--------------------: | :-------: | --------- | :-------: | :-------: | :-------: | :-------: |
> |          RIA           |   38.88   | 59.81     |   26.54   |   19.20   |   53.43   |   39.57   |
> |         RIA+CP         |   40.17   | 61.57     |   28.33   |   20.40   |   52.71   |   40.64   |
> | $\text{PermLLM}_{RIA}$ | **43.44** | **62.63** | **31.83** | **21.20** | **56.32** | **43.08** |
>
> For LLaMA 7B,
>
> |         Method         | HellaSwag | ARC_E     |   ARC_C   |   OBQA    |    RTE    |  Average  |
> | :--------------------: | :-------: | --------- | :-------: | :-------: | :-------: | :-------: |
> |          RIA           |   42.74   | 61.11     |   28.42   |   24.00   |   57.40   |   42.73   |
> |         RIA+CP         |   44.34   | 62.63     |   29.95   | **25.00** |   55.23   |   43.43   |
> | $\text{PermLLM}_{RIA}$ | **45.49** | **64.48** | **30.72** |   24.00   | **58.12** | **44.56** |
>
> For LLaMA-2 7B,
>
> |         Method         | HellaSwag | ARC_E     |   ARC_C   |   OBQA    |    RTE    |  Average  |
> | :--------------------: | :-------: | --------- | :-------: | :-------: | :-------: | :-------: |
> |          RIA           |   41.81   | 62.12     |   28.07   |   22.60   |   54.15   |   41.75   |
> |         RIA+CP         |   42.86   | 64.69     |   30.29   |   24.40   | **54.87** |   43.42   |
> | $\text{PermLLM}_{RIA}$ | **45.15** | **64.77** | **32.25** | **24.80** |   54.51   | **44.30** |
>
> For OPT 6.7B
>
> |         Method         | HellaSwag | ARC_E     |   ARC_C   |   OBQA    |    RTE    |  Average  |
> | :--------------------: | :-------: | --------- | :-------: | :-------: | :-------: | :-------: |
> |          RIA           | **42.56** | 59.55     |   26.45   |   22.60   | **54.51** |   41.13   |
> |         RIA+CP         |   44.45   | **60.44** |   26.54   | **24.80** |   51.26   |   41.50   |
> | $\text{PermLLM}_{RIA}$ |   44.27   | 59.43     | **27.22** |   24.00   |   54.15   | **41.81** |
>
> For LLaMA 13B,
>
> |          Method          | HellaSwag | ARC_E     |   ARC_C   |   OBQA    |    RTE    |  Average  |
> | :----------------------: | :-------: | --------- | :-------: | :-------: | :-------: | :-------: |
> |          Wanda           |   47.16   | 66.12     |   33.11   |   27.40   |   54.15   |   45.59   |
> |         Wanda+CP         |   49.49   | **69.99** | **36.01** | **28.40** |   52.71   |   47.32   |
> | $\text{PermLLM}_{Wanda}$ | **51.27** | 68.52     | **36.01** |   28.00   | **55.96** | **47.95** |
>
> For LLaMA-2 13B,
>
> |          Method          | HellaSwag | ARC_E     |   ARC_C   |   OBQA    |    RTE    |  Average  |
> | :----------------------: | :-------: | --------- | :-------: | :-------: | :-------: | :-------: |
> |          Wanda           |   46.77   | 68.43     |   35.75   |   24.60   |   56.68   |   46.45   |
> |         Wanda+CP         |   49.24   | 70.16     |   36.60   | **29.40** |   55.60   |   48.20   |
> | $\text{PermLLM}_{Wanda}$ | **51.88** | **70.41** | **37.46** |   28.40   | **59.93** | **49.62** |
>
> These results demonstrate that PermLLM is able to enhance pruning performance regardless of the one-shot pruning method applied.
>
> **Q3**: Yes, we will include Pruner-Zero and MaskLLM for comparison in the final version.
>
> For LLaMA 7B under 2:4 sparsity,
>
> |          Method          | HellaSwag |   ARC_E   |   ARC_C   |   OBQA    |    RTE    | Wikitext2 |
> | :----------------------: | :-------: | :-------: | :-------: | :-------: | :-------: | :-------: |
> |       Pruner-Zero        | **56.29** |   60.56   | **30.63** | **34.40** |   53.79   |   10.61   |
> | $\text{PermLLM}_{Wanda}$ |   47.03   | **63.30** |   30.55   |   25.00   | **62.45** | **9.41**  |
>
> For LLaMA-2 7B under 2:4 sparsity,
>
> |          Method          | HellaSwag |   ARC_E   |   ARC_C   |   OBQA    |    RTE    | Wikitext2 |
> | :----------------------: | :-------: | :-------: | :-------: | :-------: | :-------: | :-------: |
> |       Pruner-Zero        | **54.68** |   61.57   | **32.17** | **32.60** |   53.43   |   10.52   |
> | $\text{PermLLM}_{Wanda}$ |   46.60   | **65.49** |   31.14   |   26.20   | **63.54** | **9.39**  |
>
> As the results show, PermLLM can outperform Pruner-Zero on certain tasks even when using Wanda (which is originally much worse than Pruner-Zero) as the pruning metric. Pruner-Zero is designed to automatically search for the optimal pruning metric. Therefore, PermLLM can be integrated with Pruner-Zero, resulting in $\text{PermLLM}_{Pruner-Zero}$, which utilizes the pruning metric identified by Pruner-Zero. In this way, PermLLM can further improve the performance of Pruner-Zero.
>
> As for MaskLLM, they use a blended training set and the training overhead is significant, which is different from the experimental settings by us and the baselines we used. We implement MaskLLM using 128 samples from C4 for training, resulting in a perplexity exceeding 5k on LLaMA-3.1 8B and Qwen-2.5 7B models. This demonstrates that the performance of MaskLLM heavily depends on sufficient training data, which is consistent with the findings presented in Figure 4 of their paper. If available, one can perform PermLLM first to find a relative good channel order, which will be beneficial for subsequent mask learning by MaskLLM.
>
>
>
> Thank you once again for your valuable suggestions to enhance the quality of our paper!

---

> > ### Comment · Reviewer_fovA · 2025-08-04
> >
> > I believe the authors’ response has addressed most of my concerns regarding the experimental results, so I will raise my score to **4**. Additionally, I suggest that in Table 2, the number of models could be appropriately reduced, while including a wider variety of methods. This would make it easier to directly compare the effectiveness of different approaches. The remaining models can be moved to the appendix.

---

### Official Review · Reviewer_m4zW · 2025-07-02

**Clarity:** 2
**Significance:** 2
**Originality:** 1
**Rating:** 3
**Confidence:** 4

**Summary:**

PermLLM is a post-training pruning framework that makes *channel permutation itself learnable*. The core idea is to relax the discrete permutation matrix into a soft, doubly-stochastic matrix produced by Sinkhorn normalization, then “harden” it back to a true permutation with the Hungarian algorithm while passing gradients through a straight-through estimator. To keep the search space tractable for LLM-scale layers, the permutation is applied **block-wise**, cutting the learnable parameters from $C_{\text{in}}^{2}$ to $C_{\text{in}}B$ and reducing the Hungarian step from $O(C_{\text{in}}^{3})$ to $O(C_{\text{in}}B^{2})$. Plugged into one-shot N:M pruning schemes such as Wanda and RIA, the learnable permutation is trained to minimize a cosine loss between dense and sparse outputs instead of maximizing a handcrafted importance score. On seven open-source LLMs (OPT-6.7B, LLaMA-1/2/3.1 7–13B, Qwen-2.5 7B) the 2:4 sparse models show consistent gains: perplexity on WikiText-2 drops from 11.59 to **9.41** for LLaMA-7B (vs. Wanda) and average zero-shot accuracy rises by 3–5 pp across five tasks. A custom CUDA kernel makes permutation overhead negligible (0.039 ms) and still delivers 1.67× faster inference than the dense model. Figure 1 illustrates how a naive “max-score” permutation can actually *hurt* accuracy, motivating the learnable approach.

**Questions:**

1. How do accuracy and training time vary with block size beyond the single ablation shown? Could adaptive per-layer block sizes yield further gains?
2. Can learnable permutation be limited to the most sensitive layers (as hinted by a “partial PermLLM” experiment) without retraining dense layers and still match full performance?
3. Have the pruned models been evaluated on out-of-domain datasets or downstream fine-tuning to ensure the learned permutations do not overfit the calibration set?
4. Since many deployments combine sparsity with 4-bit or 8-bit quantization, does PermLLM remain effective when applied after GPTQ or similar schemes?
5. The custom CUDA kernel targets NVIDIA sparse tensor cores. How would the approach translate to hardware lacking these primitives (AMD MI-series, Intel Gaudi)?

**Ethical Concerns:**

["NO or VERY MINOR ethics concerns only"]

**Limitations:**

PermLLM still requires a non-trivial GPU-heavy optimization loop—unlike heuristic channel permutation it is **not** a free post-processing step. Even with block-wise sharing, the method introduces additional parameters and hyperparameters (temperature schedule, Sinkhorn iterations) that may need tuning per model. The reported speed-up (≈1.67×) is far below the theoretical 2× for 2:4 sparsity, indicating residual inefficiencies. Accuracy gains, though consistent, are sometimes modest (<2 pp) on the hardest reasoning tasks, suggesting diminishing returns at higher sparsity. The method has only been validated on decoder-only transformer LLMs; applicability to encoder–decoder or vision transformers remains to be proven.

**Quality:**

2

**Strengths And Weaknesses:**

**Strengths**
* First to cast channel permutation as a differentiable layer for N:M sparsity and integrate it with existing one-shot pruning methods, replacing brittle heuristics with end-to-end loss minimization.
* Sinkhorn–Hungarian–STE pipeline is mathematically clean, and the block-wise design balances search power and cost.
* Broad experimental coverage (7 LLMs, language modeling plus reasoning tasks) with reproducible gains over SparseGPT, Wanda-CP, and RIA-CP in both 2:4 and 4:8 sparsity.
* Practical implementation: bespoke kernel yields an 84× speed-up over PyTorch permutation and preserves overall sparsity speed-ups.

**Weaknesses**
* Training permutations adds wall-time (≈2.5 h for 7B, 5.5 h for 13B on multiple A100s), reducing the appeal of “one-shot” pruning.
* Block size is fixed at 64; larger or adaptive blocks are unexplored.
* Speed-ups are throughput-only; memory footprint and launch overhead are not discussed.
* Analysis focuses on 2:4 sparsity; 4:8 results are promising but less detailed.
* Calibration uses only 128 C4 samples, so robustness to domain shifts is unclear.

---

> ### Author Rebuttal · Authors · 2025-07-31
>
> We appreciate the time and effort the reviewer has invested in evaluating our work.
>
> **W1**: While permutation training adds wall-time, it enables significant performance improvements in pruning. Unlike naive one-shot pruning, which often sacrifices performance, PermLLM achieves a better accuracy, especially on the updated LLMs, such as LLaMA-3.1 and Qwen 2.5.
>
> **W2**: We conduct additional ablation study about block size and we provide a discussion about adaptive block, which can be found in **Q1**.
>
> **W3**: The memory overhead of PermLLM is minimal. For example, if a weight matrix $\mathbf{W} \in \mathbb{R}^{n\times n}$, PermLLM only need to store the permutation order index $\mathbf{p} \in \mathbb{R}^n$, which is derived from the learned permutation matrix $\mathbf{P}$. For LLaMA-2 7B, the additional memory of the permutation will be 16KB for each decoder block if INT16 is used for index data. Thus, the overall memory overhead introduced by PermLLM is negligible. Moreover, the kernel launch time is already accounted for in the runtime evaluation.
>
> **W4**: 2:4 sparsity is the primary sparsity pattern recommended by NVIDIA, as it strikes an excellent balance between efficiency and accuracy, making it the key focus of prior research on N:M sparsity. Consequently, our work also primarily focuses on the evaluation and application of this sparsity pattern. We provide additional experimental results on Qwen-2.5 7B under 4:8 sparsity. Due to time limitation, we plan to include more evaluations on 4:8 sparsity in the final version. As the results indicate, PermLLM enhances pruning performance under 4:8 sparsity, demonstrating its strong generalization capability. We also conduct experiments under 1:4 sparsity and 3:4 sparsity, the results can be found in the response of **Q1** by Reviewer fovA.
>
> |          Method          | HellaSwag | ARC_E     |   ARC_C   |   OBQA    |    RTE    |  Average  | Wikitext2 |
> | :----------------------: | :-------: | --------- | :-------: | :-------: | :-------: | :-------: | :-------: |
> |          Wanda           |   47.05   | **74.24** |   40.96   |   27.20   |   74.01   |   52.69   |   14.13   |
> |         Wanda+CP         |   47.76   | 73.91     |   40.61   | **28.00** |   76.53   |   53.36   |   12.98   |
> | $\text{PermLLM}_{Wanda}$ | **48.95** | 73.65     | **41.89** | **28.00** | **76.81** | **53.86** | **12.30** |
>
> **W5**: We just follow our baselines (i.e., SparseGPT, Wanda and RIA) for a fair comparison, which all use 128 samples from C4 dataset as the calibration data.
>
> **Q1**: The results about varying block size are shown as follows:
>
> | Block size | HellaSwag | ARC_E     | ARC_C     | OBQA      | RTE       | Average   | Wikitext2 |
> | ---------- | --------- | --------- | --------- | --------- | --------- | --------- | --------- |
> | 32         | 46.13     | 64.39     | 29.69     | 24.60     | 53.07     | 43.58     | 9.50      |
> | 64         | **46.60** | 65.49     | 31.14     | 26.20     | 63.54     | 46.59     | 9.39      |
> | 128        | 46.47     | **66.08** | **32.08** | **27.40** | **64.43** | **47.09** | **9.07**  |
>
> The training time is ~2h for block size=32, ~2.5h for block size=64 and ~6h for block size=128. A larger block size provides more optimization potential but also results in longer exploration and convergence times. We set the block size to 64 as the default configuration, as it strikes a good balance between pruning performance and training efficiency.
>
> Furthermore, we believe that using an adaptive, per-layer block size could lead to further improvements. By measuring the sensitivity or importance of each layer, larger block sizes could be assigned to the more critical layers, ensuring better preservation of the most important weights. However, as adaptive block size is not the primary focus of this paper, we leave it as a direction for future work.
>
> **Q2**: As shown in Table 5, partial PermLLM only inserts learnable channel permutation modules to the most sensitive layers, and applies traditional channel permutation method to the rest layers. Partial PermLLM outperforms the traditional channel permutation method but falls short compared to full PermLLM, highlighting a performance gap between the two. Because partial PermLLM leverages learnable channel permutations for the most sensitive layers, it sacrifices a significant portion of the optimization space in model-wide optimization. However, the advantage of partial PermLLM lies in its training efficiency, making it a viable option when computational resources are limited. Moreover, it is important to note that we did not perform any retraining or fine-tuning on the dense layers in any of the experiments, meaning the weight values remained unchanged and fixed. The only component we trained was the permutation matrix, which aims to identify a permutation for generating a better pruning mask for the dense layers.
>
> **Q3**: We follow the experimental settings used by our baselines (i.e., SparseGPT, Wanda and RIA) for a fair comparison. We think C4 is an out-of-domain dataset as the evaluation benchmarks are diverse and different from C4. We also conduct additional experiments using 128 samples from Pile [1] or 128 samples from Wikitext2 [2]. When using Wikitext2 as the calibration data, the pruned model performs better on Wikitext2 compared to using Pile or C4. Since we use C4 as the calibration data and apply the same dataset across all baselines, we believe the settings are fair and appropriate. And if overfitting the calibration data, we believe the results will be much worse on the evalution benchmarks than the existing results.
>
> | Dataset   | HellaSwag | ARC_E | ARC_C | OBQA  | RTE   | Average | Wikitext2 |
> | --------- | --------- | ----- | ----- | ----- | ----- | ------- | --------- |
> | Pile      | 45.83     | 64.31 | 32.08 | 26.60 | 54.87 | 44.74   | 8.96      |
> | Wikitext2 | 45.42     | 66.41 | 32.34 | 25.80 | 53.07 | 44.61   | 8.31      |
> | C4        | 46.60     | 65.49 | 31.14 | 26.20 | 63.54 | 46.59   | 9.39      |
>
> **Q4**: Yes, PermLLM remains effective when applied after 8bit GPTQ, and the results are shown as follows. Here, we perform pruning after quantization and we believe an interesting future work will be pruning- and quantization- aware permutaion learning, which can concurrently find a suitable permutation for improved pruning and quanzation effect.
>
> |            Method             | HellaSwag | ARC_E     |   ARC_C   |   OBQA    |    RTE    |  Average  | Wikitext2 |
> | :---------------------------: | :-------: | --------- | :-------: | :-------: | :-------: | :-------: | :-------: |
> |          Wanda+GPTQ           |   41.50   | 61.41     |   30.55   |   23.20   |   53.07   |   41.95   |   12.08   |
> |         Wanda+CP+GPTQ         |   43.94   | 64.52     |   30.80   |   24.00   |   53.07   |   43.27   |   10.92   |
> | $\text{PermLLM}_{Wanda}$+GPTQ | **46.42** | **65.99** | **31.83** | **27.20** | **53.43** | **44.97** | **9.04**  |
>
> **Q5**: Since NVIDIA GPUs are the dominant platform for research, our efforts primarily focus on developing custom kernels optimized for NVIDIA GPUs, as has been the case with most previous work.  Just as unstructured sparsity does not achieve significant acceleration on GPUs, this has not stopped researchers from extensively exploring it. Similarly, N:M sparsity was initially proposed by NVIDIA, so it is reasonable that it achieves better adaptation on NVIDIA GPUs. Even if acceleration is currently limited to NVIDIA GPUs, there is still a great deal of ongoing research focused on optimizing N:M sparsity.
>
> The permutation kernel we developed is not strictly limited to NVIDIA GPUs, as the operation itself is essentially just reordering a matrix. If hardware such as AMD MI-series or Intel Gaudi supports parallel operations, we can adapt the kernel accordingly to make it compatible with those platforms, extending the applicability of our approach.
>
> **L1**: "The reported speed-up (≈1.67×) is far below the theoretical 2× for 2:4 sparsity, indicating residual inefficiencies." $2\times$ speedup is theoretical, and it does not mean everyone can achieve under any cases. SparseGPT and RIA also report speedup under 2:4 sparsity, which can not achieve the theoretical speedup either. Thus, this is not our limitation; it is a limitation of the hardware itself, as well as the gap between theoretical analysis and practical application. The only limitation about runtime is the additional channel permutation operations we introduce. With the developed CUDA kernel, the overhead is minimal.
>
> **L2**: Our methodology is not restricted to decoder-only transformers. As demonstrated in the paper, it does not rely on any specific components exclusive to LLM. We will open source our framework so that other researchers or users are able to extend it to any architectures. Furthermore, due to the structural similarities across model architectures used by encoder-decoder models or vision transformers, it can be easily adapted for them.
>
>
>
> Thanks again for your valuable suggestions to improve the quality of our paper!
>
>
>
> References
>
> [1] Gao, Leo, et al. "The pile: An 800gb dataset of diverse text for language modeling." *arXiv preprint arXiv:2101.00027* (2020).
>
> [2] Merity, Stephen, et al. "Pointer sentinel mixture models." *arXiv preprint arXiv:1609.07843* (2016).

---

### Official Review · Reviewer_gs8N · 2025-07-02

**Clarity:** 3
**Significance:** 2
**Originality:** 3
**Rating:** 4
**Confidence:** 2

**Summary:**

PermLLM addresses the limitations of heuristic-driven channel permutation in N:M sparse LLMs by proposing a novel learnable channel permutation (LCP) framework that optimizes permutations end-to-end to minimize pruning-induced output errors. As a plug-in enhancement to existing one-shot methods (e.g., Wanda), PermLLM achieves outstanding results across LLaMA, OPT, and Qwen families—reducing perplexity degradation and improving zero-shot accuracy.

**Questions:**

How robust is the learned permutation to the choice and size of the calibration dataset? Does performance vary significantly with different calibration data?

**Ethical Concerns:**

["NO or VERY MINOR ethics concerns only"]

**Limitations:**

Yes.

**Paper Formatting Concerns:**

No formatting concerns.

**Quality:**

2

**Strengths And Weaknesses:**

### Strengths

The paper compellingly argues against handcrafted metrics (Fig. 1) and introduces a novel framework for learnable channel permutation (LCP) that directly minimizes pruning error, which is an original contribution to semi-structured pruning.

The block-wise permutation strategy is a clever solution to make LCP scalable for LLMs, reducing parameter count and conputational complexity.

The method is thoroughly evaluated on a wide range of modern LLMs, mostly outperforming baselines in both perplexity and zero-shot tasks.


### Weaknesses

The choice of block size (64) is justified anecdotally rather than empirically.

 An ablation study showing the trade-off between block size, performance, and training cost is missing.

The experimental results are not optimal in every setting.

---

> ### Author Rebuttal · Authors · 2025-07-31
>
> Thank you for taking the time and effort to review our paper.
>
> **W1**: We conduct additional experiments about $\text{PermLLM}_{Wanda}$ on LLaMA-2 7B to show the trade-off between performance and training cost with varying block size.
>
> | Block size | HellaSwag | ARC_E     | ARC_C     | OBQA      | RTE       | Average   | Wikitext2 |
> | ---------- | --------- | --------- | --------- | --------- | --------- | --------- | --------- |
> | 32         | 46.13     | 64.39     | 29.69     | 24.60     | 53.07     | 43.58     | 9.50      |
> | 64         | **46.60** | 65.49     | 31.14     | 26.20     | 63.54     | 46.59     | 9.39      |
> | 128        | 46.47     | **66.08** | **32.08** | **27.40** | **64.43** | **47.09** | **9.07**  |
>
> The training time is ~2h for block size=32, ~2.5h for block size=64 and ~6h for block size=128. A larger block size offers a greater optimization space. However, this increased space leads to longer exploration and convergence times. We choose block size=64 as the default configuration because it can achieve good balance between pruning performance and training efficiency.
>
> **W2**: As the benchmarks we used in the evaluation are diverse, so it is extremely challenging  to achieve optimal results in every benchamark. Thus, we believe the evaluation results are good enough to demonstrate the superiority of the proposed PermLLM.
>
> **Q1**: We perform extra experiments about $\text{PermLLM}_{Wanda}$ on LLaMA-2 7B with different calibration data: (a) 128 samples from Pile[1], (b) 128 samples from Wikitext2 [2]and (c) 128 samples from C4 [3]. The learned permutation performs consistently well across different datasets, it indicates robustness of PermLLM.
>
> | Dataset   | HellaSwag | ARC_E | ARC_C | OBQA  | RTE   | Average | Wikitext2 |
> | --------- | --------- | ----- | ----- | ----- | ----- | ------- | --------- |
> | Pile      | 45.83     | 64.31 | 32.08 | 26.60 | 54.87 | 44.74   | 8.96      |
> | Wikitext2 | 45.42     | 66.41 | 32.34 | 25.80 | 53.07 | 44.61   | 8.31      |
> | C4        | 46.60     | 65.49 | 31.14 | 26.20 | 63.54 | 46.59   | 9.39      |
>
> We also use different size of calibration dataset (C4) to check the performance of PermLLM.
>
> | Size | HellaSwag | ARC_E | ARC_C | OBQA  | RTE   | Average | Wikitext2 |
> | ---- | --------- | ----- | ----- | ----- | ----- | ------- | --------- |
> | 32   | 46.08     | 65.11 | 31.66 | 25.40 | 53.13 | 44.28   | 9.60      |
> | 64   | 46.61     | 64.14 | 32.00 | 25.20 | 53.43 | 44.28   | 9.46      |
> | 128  | 46.60     | 65.49 | 31.14 | 26.20 | 63.54 | 46.59   | 9.39      |
>
> The performance of PermLLM improve gradually with increased size of calibration data, which also demonstrate the robustness of the proposed methods.
>
>
>
> Thank you again for your helpful advice!
>
>
>
> References
>
> [1] Gao, Leo, et al. "The pile: An 800gb dataset of diverse text for language modeling." *arXiv preprint arXiv:2101.00027* (2020).
>
> [2] Merity, Stephen, et al. "Pointer sentinel mixture models." *arXiv preprint arXiv:1609.07843* (2016).
>
> [3] Raffel, Colin, et al. "Exploring the limits of transfer learning with a unified text-to-text transformer." *Journal of machine learning research* 21.140 (2020): 1-67.

---

> > ### Comment · Reviewer_gs8N · 2025-08-05
> > **Official Comment by Reviewer gs8N**
> >
> > Thank you for the thoughtful response. The newly provided experiments and explanations have satisfactorily resolved the issues I previously raised. I have no additional concerns at this time.

---

### Official Review · Reviewer_uQ84 · 2025-07-03

**Clarity:** 4
**Significance:** 3
**Originality:** 2
**Rating:** 5
**Confidence:** 5

**Summary:**

Channel permutations have been shown to work well at reducing the lost weight magnitude or other particular metric, but these metrics do not always maximize model quality.  Further, existing work uses approximate heuristics or computationally expensive search procedures to find good permutations for N:M sparsity.  PermLLM addresses these two shortcomings by directly learning the permutations.  First, a standard permutation matrices are relaxed into "soft" permutation matrices in order to make them differentiable, using Sinkhorn normalization to convert the learned matrix into a doubly stochastic matrix, which can be hardened into a true permutation for forward propagation.  To reduce the complexity and number of learnable parameters, the permutation matrices are simplified into block-wise permutation matrices.  A soft mask is used in the backwards pass to allow differentiability when learning the permutations (since they will change the mask applied), and cosine similarity between the dense and sparse model outputs is used as the loss function.  Reasonably broad evaluations are performed, showing meaningful average improvements over the considered baselines, and top-quality results on most model/task pairs.

**Questions:**

1. How does PermLLM compare to previous work (say, [42]) operating with a better importance metric than magnitude (say, Wanda's importance metric)?
2. How does PermLLM compare to learned rotations (RotPruner)?  Is there room for both techniques?
3. How does PermLLM compare to past approaches to directly learning permutations (2, above)
4. How does PermLLM perform with varying block sizes (32, 64, 128 (if possible!))?

Addressing these questions will directly affect the quality, originality, and (most importantly) the significance of the submission.  Results on a single model should be enough to show the important trends.

**Ethical Concerns:**

["NO or VERY MINOR ethics concerns only"]

**Final Justification:**

PermLLM builds on a solid foundation of past work, but is a novel advancement that meaningfully improves N:M model quality with relatively low overhead.  The authors satisfactorily addressed all of my concerns.

**Limitations:**

Limitations are discussed in the appendix, but should really be in the main content.

**Quality:**

3

**Strengths And Weaknesses:**

## Strengths
The idea is well-motivated; figure 1 clearly shows how existing metrics are not ideal to maximize model quality.  The approach is also sound; each piece of the solution is clearly motivated and explained, and the effect of the whole process is proven successful by the evaluations.  I also appreciate the runtime analysis, as well as the ablation study (though it is somewhat limited).  I have little doubt that a dedicated practitioner could reproduce the results from studying the descriptions and equations, and that the reported model quality improvements would likely generalize beyond the presented results.

## Weaknesses
I want to believe in this submission's superiority over past work, but there are some missed references and baselines that affect the significance, originality, and quality of the submission.

Some missing references should be included:
- Mahajan et al. [1] formulate the permutation search as a discrete optimization problem.
- Lyu et al. [2] learn permutation matrices directly.  This one is particularly relevant, and the authors should provide a comparison of this approach and the proposed approach, and, ideally, empirical comparisons.

A simple baseline would be the approach from [42] applied to e.g. SparseGPT scores or Wanda importance metrics (as noted in that work, the technique is applicable to any metric, not just magnitude).  The authors note on line 45 that the large search space renders this approach infeasible for large sizes, but [42] includes less-expensive settings than the default.  Using the simplest from [42]'s Table 3 and the available code on a single A100, searching for a good permutation for a random 4096x4096 matrix (such as Llama2 7B's attention output projection) took only 125 seconds.  I continued timing random matrices for the other layers and came up with these results:

| Layer | Size | Duration (s)|
| ------ | ------ | -------|
| QKV | 4096x12288 | 297 |
| PROJ | 4096x4096 | 125 |
| up| 4096x11008 | 268 |
| gate | 4096x11008 | 268 |
| down | 11008x4096 | 1318 |

Replicated for 32 blocks, the total projected time to search for permutations for all of Llama2 7B is 20.25 hours.  If parallelized over 4 GPUs to match the submission's resources for this model, this is "only" double the 2.5 hours reported on line 298.  It may indeed be double the cost, but certainly not "prohibitively high computation complexity" as claimed on line 46.  If, as suggested above, these permutations are found using a higher-quality importance metric than weight magnitude, they could prove competitive with PermLLM.  Or, they may not.  I believe this is a useful baseline to correctly determine the significance of PermLLM as a technique as well as improve the quality of the submission.

I'm also concerned about the interaction with another, non-orthogonal technique that seeks to reduce the quality impact of N:M sparsity: rotations.  They've been successfully used for quantization [3,4] as well as N:M sparsity[5].  It's not clear that PermLLM composes with or outperforms this technique.

Finally, some small issues reduce the clarity and quality of the submission:
- On line 269, the authors suggest that the additional permutation to handle residual paths has a negligible impact on deployment efficiency, citing [58].  In the next sentence, though, the authors describe a custom CUDA kernel to reduce the overhead.  Why do we care about reducing a negligible overhead?  (The experimental results on line 271 probably belong in Section 5, too.)
- Sweeping the number of iterations used in Sinkhorn normalization in Section 5.3 would be informative.  (5 is better than 0, but is 5 enough?  More than enough?)
- An ablation of the block size used (32, 64, 128) would be useful to see how sensitive the outcome is to this hyperparameter.
- What software is used to gather the runtimes in Table 3?

References
1. Mahajan, M., Hwu, WM. & Nagi, R. Determining optimal channel partition for 2:4 fine grained structured sparsity. Optim Lett 18, 2079–2090 (2024). https://doi.org/10.1007/s11590-023-02084-8
2. Jiancheng Lyu, Shuai Zhang, Yingyong Qi, and Jack Xin. AutoShuffleNet: Learning Permutation Matrices via an Exact Lipschitz Continuous Penalty in Deep Convolutional Neural Networks. In Proceedings of the 26th ACM SIGKDD International Conference on Knowledge Discovery & Data Mining (KDD '20). https://doi.org/10.1145/3394486.3403103
3. Saleh Ashkboos, Amirkeivan Mohtashami, Maximilian L. Croci, Bo Li, Pashmina Cameron, Martin Jaggi, Dan Alistarh, Torsten Hoefler, James Hensman, QuaRot: Outlier-Free 4-Bit Inference in Rotated LLMs, NeurIPS 2024
4. Zechun Liu, Changsheng Zhao, Igor Fedorov, Bilge Soran, Dhruv Choudhary, Raghuraman Krishnamoorthi, Vikas Chandra, Yuandong Tian, Tijmen Blankevoort, SpinQuant: LLM Quantization with Learned Rotations, ICLR 2025
5. Anonymous, ROTPRUNER: LARGE LANGUAGE MODEL PRUNING IN ROTATED SPACE  https://openreview.net/pdf?id=wV9iMiyQcc

---

> ### Author Rebuttal · Authors · 2025-07-31
>
> Thanks for your detailed review and valuable feedback.
>
> **W1**: Thanks for your reminding. We will include the missing references and discuss them in the final version.
>
> (a) Mahajan et al. [1] and [42] propose different methods to achieve a good permutation result for N:M sparsity. However, their major limitations lie in the biased optimization objective (focusing on maximum retained weight importance rather than the output loss between the original model and the pruned model) and the local optimality of the permutation matrix (performing layer-wise optimization, which fails to capture inter-layer dependencies). These issues, as highlighted in the motivation section, limit their ability to further improve performance. To further validate this, we implement the method from [42] combined with Wanda for a direct comparison against the proposed PermLLM. Please refer to **Q1** for detailed results. "Prohibitively high computational complexity" in line 46 refers to the solution space complexity of concurrently finding permutations for all layers. For example, if we need to determine two permutations for two linear layers in the model, and there are n possible permutations for layer 1 and m for layer 2, the complexity is n+m when performing layer-wise optimization. However, for model-wise optimization, the complexity increases to n*m. We will rewrite it to avoid misunderstanding.
>
> (b) Lyu et al. [2] employ an $\ell_{1-2}$ penalty to constrain the learnable matrix, ensuring that, upon convergence, each row and column contains exactly one entry equal to 1, with all other entries being zero. They also utilize Sinkhorn normalization to facilitate convergence. The key difference between their approach and ours lies in the method used to achieve convergence to 1 or 0, which we think it is minor: while they rely on the $\ell_{1-2}$ penalty, we adopt an annealing-based method. We believe that two approaches can achieve similar results with appropriate training configurations. Furthermore, the more valuable contributions of PermLLM are its abilities to achieve pruning-aware permutation learning for the first time and efficiently train the permutation matrix, which provides a new pruning framework for further exploration.
>
> **W2**: RotPruner [5] prunes the model in rotated space. As shown in Fig.2 of their paper, the rotation operation results in more values being 0 or small magnitude. We think it may be more benefical for unstructured pruning as N:M sparisty has more strict constraints.
>
> There are some shortcomings compared with PermLLM: (a) memory overhead: RotPrune requires storing the rotation matrices in memory. For a weight matrix $\mathbf{W} \in \mathbb{R}^{n \times n}$, the corresponding rotation matrix $\mathbf{Q} \in \mathbb{R}^{n \times n}$ has the same size, leading to significant memory usage. In contrast, PermLLM only needs to store a vector $\mathbf{v} \in \mathbb{R}^{n}$ to represent the column permutation order, resulting in much lower memory overhead. (b) runtime overhead: RotPrune requires online computation of rotations, which is less efficient compared to the lightweight permutation operation. Rotation can be seen as an additional dense linear layer, introducing extra computational costs (e.g., Q/K/V_proj vs. CP) in Table 3.
>
> PermLLM can also outperform RotPruner, the results can be found in **Q2**.
>
> **W3**: We will revise the paragraph in the final version for improved clarity. The original PyTorch implementation is inefficient, so we develop a custom CUDA kernel to achieve significant speedup. With this optimized kernel, the additional permutation incurs minimal overhead, resulting in a negligible overall impact. Thank you for bringing this to our attention!
>
> **W4**: We conduct additional experiments on LLaMA-2 7B by varying the number of iterations used in Sinkhorn normalization. With the exception of Wikitext2, iter=5 outperforms both iter=0 and iter=10 on the remaining benchmarks. The reason why iter=5 performs better is that introducing some perturbations during the early exploration helps in finding a better solution and avoiding being trapped in a local optimum.
>
> | Iter. | HellaSwag | ARC_E     | ARC_C     | OBQA      | RTE       | Average   | Wikitext2 |
> | ----- | --------- | --------- | --------- | --------- | --------- | --------- | --------- |
> | 0     | 46.23     | 64.48     | 30.80     | 25.00     | 54.51     | 44.20     | 9.15      |
> | 5     | **46.60** | **65.49** | **31.14** | **26.20** | **63.54** | **46.59** | 9.39      |
> | 10    | 46.37     | 63.17     | 28.50     | 25.80     | 55.23     | 43.81     | **9.07**  |
>
> **W5**: Block size is a critical hyperparameter that balances pruning performance and training efficiency. To explore this trade-off, we conduct additional experiments by varying the block size, which can be found in **Q4**.
>
> **W6**: We used SparseSemiStructuredTensor and to_sparse_semi_structured in Pytorch and enabled CUTLASS for sparse operations. Moreover, we utilized Timer utility from torch.utils.benchmark to measure runtimes.
>
> **Q1**: We apply [42] with Wanda to prune LLaMA-2 7B. The results below demonstrate the superiority of PermLLM. There is no doubt that our motivation is well-founded and our methodology highly effective. The reasons behind PermLLM's superior performance have been discussed in **W1**.
>
> |          Method          | HellaSwag |   ARC_E   |   ARC_C   |   OBQA    |    RTE    |  Average  | Wikitext2 |
> | :----------------------: | :-------: | :-------: | :-------: | :-------: | :-------: | :-------: | :-------: |
> |        [42]+Wanda        |   43.41   |   62.92   |   28.67   | **26.20** |   54.51   |   43.14   |   10.19   |
> | $\text{PermLLM}_{Wanda}$ | **46.60** | **65.49** | **31.14** | **26.20** | **63.54** | **46.59** | **9.39**  |
>
> **Q2**: In RotPruner, 128 samples from Wikitext2 are used as calibration data. Thus, we also use 128 samples in Wikitext2 as calibration data for PermLLM. The results are shown below:
>
> |             Method             | HellaSwag | ARC_E | ARC_C | OBQA  |  RTE  | Average | Wikitext2 |
> | :----------------------------: | :-------: | :---: | :---: | :---: | :---: | :-----: | :-------: |
> |        RotPruner (2:4)         |     -     |   -   |   -   |   -   |   -   |    -    |   9.20    |
> | $\text{PermLLM}_{Wanda}$ (2:4) |   45.42   | 66.41 | 32.34 | 25.80 | 53.07 |  44.61  | **8.31**  |
>
> We didn't find the results for the zero-shot tasks under 2:4 sparsity in their paper. However, for Wikitext-2, PermLLM significantly outperforms RotPruner. Moreover, we believe there is room for both techniques, making it an interesting avenue for further exploration and improvement.
>
> For example, $\mathbf{x}$ = [[-1.0412],[-1.4108],[ 1.0199],[-1.7633],[-0.2013],[ 0.8784],[ 0.2520],[ 1.3273]],
>
> $\mathbf{W}$=[[ 0.5143,  0.5205,  0.6937,  0.0350, -0.4507,  0.3604,  2.4873,  0.9460],[ 0.0755, -0.9656,  1.2620, -0.7471, -2.5672,  1.6677, -1.1052,  0.6418]],
>
> rotation matrix $\mathbf{Q}$=[[-0.1583, -0.5283,  0.4140, -0.0586,  0.2760, -0.0372,  0.1789, -0.6414],
>         [ 0.6179, -0.2853, -0.1028, -0.5905, -0.0206,  0.2548, -0.3318, -0.0461],
>         [-0.2811, -0.3364, -0.3383, -0.2190,  0.4878,  0.2649,  0.3716,  0.4463],
>         [ 0.1433,  0.3674, -0.1850, -0.4794, -0.0594, -0.3628,  0.6207, -0.2449],
>         [ 0.0564, -0.2713, -0.7957,  0.3407, -0.0890, -0.1264, -0.0599, -0.3829],
>         [-0.2781, -0.3765,  0.0066, -0.2865, -0.2223, -0.7095, -0.2565,  0.2831],
>         [-0.3605, -0.1679, -0.0178, -0.1824, -0.7537,  0.4435,  0.1991, -0.0619],
>         [ 0.5321, -0.3844,  0.1894,  0.3723, -0.2377, -0.1262,  0.4749,  0.3110]],
>
>  permutation order=[3, 1, 5, 4, 6, 0, 2, 7]. With magnitude pruning, direct 2:4 pruning loss=1.660, rotated 2:4 pruning loss=0.769, rotated and permutation 2:4 pruning loss=0.497. This example highlights the potential of combining rotation and permutation for more effective N:M pruning.
>
> **Q3**: Based on our analysis in **W1**, the differences between [2] and permutation learning in PermLLM are minimal. Consequently, with proper usage, the final performance is expected to be similar. We believe this aspect is not particularly critical for the evaluation of PermLLM, as, similar to mask learning in prior pruning works, there are various methods to learn the pruning mask, such as SR-STE [i], decaying-based methods [ii], and others. The more valuable contributions lie in implementing permutation learning to successfully enhance pruning performance and make this approach more practical—achievements that PermLLM fully realizes. At the same time, we hope that our motivation and approach can inspire more researchers to further advance this field. For example, it could be applied to quantization, combined with rotation as you mentioned, or lead to the development of better pruning-aware permutation learning framework.
>
> **Q4**: We conduct an ablation study about block size to see its impact on final pruning performance and training cost.
>
> | Block size | HellaSwag | ARC_E     | ARC_C     | OBQA      | RTE       | Average   | Wikitext2 |
> | ---------- | --------- | --------- | --------- | --------- | --------- | --------- | --------- |
> | 32         | 46.13     | 64.39     | 29.69     | 24.60     | 53.07     | 43.58     | 9.50      |
> | 64         | **46.60** | 65.49     | 31.14     | 26.20     | 63.54     | 46.59     | 9.39      |
> | 128        | 46.47     | **66.08** | **32.08** | **27.40** | **64.43** | **47.09** | **9.07**  |
>
> The training time is ~2h for block size=32, ~2.5h for block size=64 and ~6h for block size=128. A larger block size offers a greater optimization space. However, this increased space leads to longer exploration and convergence times.
>
>
>
> Thank you again for your constructive suggestions!
>
>
>
> References
>
> [i] Zhou et al., LEARNING N:M FINE-GRAINED STRUCTURED SPARSE NEURAL NETWORKS FROM SCRATCH, ICLR 2021.
>
> [ii] Kao et al., Training Recipe for N:M Structured Sparsity with Decaying Pruning Mask, ICML2021 workshop.

---

> > ### Comment · Reviewer_uQ84 · 2025-08-04
> >
> > Thank you for your responses - answers to my questions 1, 2, and 4 were entirely adequate.  However, I'm struggling to understand the differences between this submission and [2].  Could you provide a precise description of what is common between the two works and what is novel to PermLLM and your submission?
> >
> > Second, one of the main motivations of your work is that optimizing permutations for magnitude preservation can be counter-productive to model quality.  However, the only data to support this motivation is from what may be a contrived example in Figure 1.  Is there any other data in the submission to support this motivation?

---

> > > ### Author Response · Authors · 2025-08-04
> > >
> > > Thank you for giving us the opportunity to further address your concerns!
> > >
> > > (a) The common part between [2] and our work is that we both use a soft permutation obtained by Sinkhorn normalization during the training. The differences include: 1) hardness control: [2] uses an $\ell_{1-2}$ penalty and we uses an anneal-based method (temperature $\tau$) 2) To achieve pruning-aware permutation learning, we will harden the soft permutation during the forward pass, but [2] will use the soft form all the time and round it in the end. 3) granularity: [2] uses a full matrix- learnable permutation, but we use block-wise learnable permutation. Why we use block-wise one is that we found the computational overhead is significant in pruning-aware permutation learning. 4) task: [2] use learnable permutation for improving channel shuffling in ShuffleNet architectures, but we use learnable permutation for N:M sparsity. In [2], after applying the learnable permutation, the permuted input will be directly fed into convolution module. In our work, the permuted weight need to be further interacted with one-shot pruning metric to demetermine the mask then prune the permuted weight. We believe it's a much more complex process compared to the senario described in [2].
> > >
> > > Thus, we believe the major contribution of us is how to adapt permutation learning to pruning problem and how to make it relatively efficient. This can be analogous to learnable rotation for quantization, e.g., [i].
> > >
> > > (b) Figure 1 is the motivated example. Here, what we want to emphasize is that the traditional channel permutation methods will lead to sub-optimal solution due to the biased optimization objective. Because this example happens to perform worse than direct 2:4 pruning, we state in the last sentence of the caption that doing so "may" lead to performance degradation. However, we are not saying that all pruning with traditional channel permutations will perform worse than direct pruning. We will rewrite this section to reduce misunderstandings, thank you for pointing it out. Based on the comprehensive evaluation presented in the paper, along with additional experiments provided in the rebuttal, we believe the suboptimality of traditional channel permutation methods has been thoroughly demonstrated.
> > >
> > > If you have any other concerns, we are willing to discuss with you. Thank you once again.
> > >
> > > Reference
> > >
> > > [i] Liu, Zechun, et al. SpinQuant: LLM Quantization with Learned Rotations. ICLR2025.

---

> > > > ### Comment · Reviewer_uQ84 · 2025-08-04
> > > >
> > > > Thank you for the quick response!  I now have a clear picture of the difference between the submission and past work.
> > > >
> > > >
> > > > > Here, what we want to emphasize is that the traditional channel permutation methods will lead to sub-optimal solution due to the biased optimization objective.
> > > >
> > > > > ...
> > > >
> > > > > Based on the comprehensive evaluation presented in the paper, along with additional experiments provided in the rebuttal, we believe the suboptimality of traditional channel permutation methods has been thoroughly demonstrated.
> > > >
> > > > I must challenge this.  I do believe that you have convincingly shown that *learning* permutations, rather than searching for them via heuristic, is superior.  However, I have not seen evidence that traditional methods' optimization objective is sub-optimal *in practice*.  (Again, I believe you in theory, but your claims are not yet empirically supported.)
> > > >
> > > > What *would* show this?  Since you've already applied [42] to Llama2-7B with Wanda in your initial rebuttal, applying [42] to magnitude, as the authors originally designed the technique, should show any difference.  If the magnitude-based optimization objective is indeed sub-optimal, then worse model quality compared to [42]+Wanda will support your hypothesis.  (Including the dense baseline and un-permuted but 2:4 sparse model quality in the table would also be helpful.)

---

> > > > > ### Author Response · Authors · 2025-08-06
> > > > >
> > > > > Thanks for your reply!
> > > > >
> > > > > Here, we list the results of [42]+magntidue on LLaMA-2 7B under 2:4 sparsity (we **highlight** the better results between [42]+magnitude and [42]+Wanda). The results demonstrate that [42]+Wanda significantly outperforms [42]+magnitude on Wikitext2 and achieve a better average accuracy on zero-shot tasks.
> > > > >
> > > > > |     Method     | HellaSwag |   ARC_E   |   ARC_C   |   OBQA    |    RTE    |  Average  | Wikitext2 |
> > > > > | :------------: | :-------: | :-------: | :-------: | :-------: | :-------: | :-------: | :-------: |
> > > > > |     Dense      |   57.13   |   76.30   |   43.26   |   31.60   |   62.45   |   54.15   |   5.47    |
> > > > > |   Magnitude    |   45.41   |   61.99   |   30.20   |   21.80   |   52.35   |   42.35   |   37.94   |
> > > > > | [42]+Magnitude | **45.11** |   60.48   | **30.80** |   25.40   |   53.07   |   42.97   |   57.65   |
> > > > > |   [42]+Wanda   |   43.41   | **62.92** |   28.67   | **26.20** | **54.51** | **43.14** | **10.19** |
> > > > >
> > > > > For wikitext2, [42]+Mangitude even underperforms Magnitude on wikitext2, which may also demonstrate that traditional channel permutation will lead to performance degradation.
> > > > >
> > > > > Thank you again for your valuable suggestions! We will revise our manuscript in the final version to enhance clarity.

---

> > > > > > ### Comment · Reviewer_uQ84 · 2025-08-06
> > > > > >
> > > > > > Thank you, authors, for the continued follow-up.  I now believe you have data that supports all of your claims and shows that PermLLM is superior to straightforward extensions of past work.  I'll update my score to 5 - accept.

---

### Note · Authors · 2025-08-12

We sincerely thank the reviewers for their constructive suggestions, which have greatly helped improve the quality of our work.

Below, we summarize the major concerns raised by the reviewers to assist AC in the decision-making process (**Some questions from reviewer m4zW are also included, and the corresponding answers are acknowledged by other reviewers.**):

**Ablation on Block Size**: The selection of block size is crucial in our pruning framework. As demonstrated by our additional results, choosing an appropriate block size is vital for balancing pruning performance and training efficiency. A smaller block size restricts the optimization space, resulting in suboptimal pruning performance. Conversely, a larger block size requires significantly more time to explore the vast solution space, delaying convergence.

**Beyond 2:4 Sparsity**: While our primary experiments focus on 2:4 sparsity, we have also extended PermLLM to 4:8 sparsity in the appendix. In response to reviewer curiosity about the effectiveness of PermLLM under other sparsity patterns, we conducted additional experiments on 1:4 sparsity, 3:4 sparsity, and 4:8 sparsity. The results confirm that **PermLLM** **is not limited to 2:4 sparsity** but performs well across diverse sparsity patterns, showcasing its generalization capability.

**Combination with Other Techniques**: Our evaluation results indicate that PermLLM remains effective when integrated with quantization and outperforms rotation-based methods. **This opens up exciting avenues for future work**, such as applying learnable channel permutations to simultaneously enhance both pruning and quantization performance, or combining PermLLM with rotation-based methods to further improve pruning accuracy.

**Calibration Data**: We conducted additional ablation studies to analyze the impact of different calibration datasets and varying calibration data sizes. Our findings show that the learned permutation performs consistently well across different calibration datasets. Furthermore, increasing the size of the calibration data leads to gradual performance improvements for PermLLM, aligning with the intuition that more data helps the pruned model better align with the original dense model. Importantly, we did not observe any overfitting issues. We used a standard calibration dataset, C4, which is not part of the evaluation benchmarks. If overfitting were present, the evaluation results would have significantly deteriorated.

---

### Decision · Program_Chairs · 2025-09-17

**Decision:**

Accept (poster)

**Comment:**

The paper proposes PermLLM, a novel post-training pruning framework that introduces Learnable Channel Permutation (LCP) for achieving N:M sparsity. The method first learns a soft permutation matrix—specifically, a doubly stochastic matrix—and then refines it into a discrete permutation (as described in Equation 6). This results in an end-to-end trainable pipeline. Additionally, the authors introduce block-wise weight learning, which further reduces training overhead. The paper presents experiments across multiple models to demonstrate the effectiveness of the approach.

All reviewers responded positively to the core idea. Initial concerns focused on the need for more clarification and additional experimental results. During the rebuttal phase, the authors addressed many of these concerns, and the discussion reflected meaningful engagement. Reviewers acknowledged the improvements and expressed increased confidence in the submission.

I recommend acceptance. The proposed framework is technically sound and well-motivated, and the authors have made a good effort to clarify and strengthen the paper during the review process. For the camera-ready version, the authors should include the additional results and explanations discussed during the rebutta